# Does Total Playing Time Affect the Neuromuscular, Physiological, and Subjective Recovery of Futsal Players during a Congested Period?

**DOI:** 10.3390/sports12050139

**Published:** 2024-05-20

**Authors:** Konstantinos Spyrou, María L. Pérez Armendáriz, Pedro E. Alcaraz, Rubén Herrero Carrasco, M. A. Sajith Udayanga, Tomás T. Freitas

**Affiliations:** 1UCAM Research Center for High Performance Sport, UCAM Universidad Católica de Murcia, 30107 Murcia, Spain; mlperez4@alu.ucam.edu (M.L.P.A.); palcaraz@ucam.edu (P.E.A.); tfreitas@ucam.edu (T.T.F.); 2Facultad de Deporte, UCAM Universidad Católica de Murcia, 30830 Murcia, Spain; 3Strength and Conditioning Society, 30008 Murcia, Spain; 4Universidad Internacional Isabel I de Castilla, 09003 Burgos, Spain; rubenhffrm@gmail.com; 5Department of Sports Science, University of Sri Jayewardenepura, Colombo 10250, Sri Lanka; sajithudayanga@sjp.ac.lk; 6NAR Nucleus of High Performance in Sport, São Paulo 04753-070, Brazil

**Keywords:** five-a-side soccer, performance, team sports

## Abstract

The aims of this study were to analyze the effects of a congested period (three games in four days) on countermovement (CMJ) jump-landing metrics, heart rate variability (HRV), and total recovery quality (TQR) score in under-19 male futsal players, and to detect the differences between those who played for more minutes (HIGH_MIN_) and less minutes (LOW_MIN_). Fourteen youth futsal players (age: 17.5 ± 0.5 years; body mass: 70.2 ± 8.5 kg; height: 1.80 ± 0.1 m) participated. HRV, TQR questionnaire, and CMJ metrics (i.e., CMJ height, relative peak power (PP_REL_), eccentric and concentric impulse, braking time, and time to peak force) were registered. A linear mixed model and effect sizes (ESs) were used to assess the differences between groups and days. Considering the total sample, a significant decrease was found in the PP_REL_ and TQR score (*p* = 0.001–0.013 and ES = 0.28–0.99) on Days 2, 3, and 4 when compared to Day 1. HIGH_MIN_ group presented a significant decrease in PP_REL_ on Day 3 (*p* = 0.004; ES: 0.62; 95% CI: 0.39–2.65) when compared to Day 1, and in the TRQ score on Day 3 (*p* = 0.002; ES: 1.98; 95% CI: 0.18–2.46) and 4 (*p* = 0.003; ES: 2.25; 95% CI: 0.52–3.38) when compared to Day 1. Non-significant differences were found for the rest of the metrics and in the group LOW_MIN_. In summary, neuromuscular performance (i.e., CMJ PP_REL_) and subjective recovery were impaired in players with higher playing minutes during a match-congested period when compared to those with less on-court time.

## 1. Introduction

Futsal is characterized as a high-intensity intermittent sport, with high physical match-play demands that involve aerobic–anaerobic (e.g., walking [0–6 km·h^−1^]), low- [6.1–12 km·h^−1^], medium- [12.1–18 km·h^−1^], and high-intensity running [>18 km·h^−1^]) and neuromuscular (e.g., accelerations [>2 m·s^−2^], decelerations [>−2 m·s^−2^], repetitive high-intensity sprints, and changes of direction) systems [1,2]. In professional futsal, players have been shown to cover ~3750 m, from which ~135 m are performed at high-intensity running velocities (>18 km·h^−1^), and to complete ~5 accelerations and ~5 decelerations per min of “court time” [1]. Considering the match demands in youth futsal, a recent study by Ohmuro et al. [3] confirmed the high-intensity nature of the sport at earlier ages, despite reporting that young players covered significant lower total distance per minute (131 vs. 141 m·min^−1^) and performed less high-intensity actions (37% vs. 43%) when compared to top-level competitors.

In different team sports (e.g., soccer and basketball), players regularly face congested periods during the season, in which they are exposed to a considerable number of matches within a few days [4,5,6]. These congested periods could decrease match performance (i.e., jumping, high-intensity efforts) [7,8], increase injury risk [9,10], affect the match result [11], and negatively influence several physical capacities, such as sprinting, agility, and jumping [12]. Still, when it comes to futsal, the literature on the effects of congested periods is scarce [13,14,15,16]. In a study with elite players, Ribeiro et al. [13] found that congested fixtures did not negatively affect physical match performance (i.e., total distance covered, high-speed running, sprinting, accelerations and decelerations), but internal (i.e., rate of perceived exertion (RPE)) and external load metrics (i.e., total distance covered, high-speed running, and accelerations and decelerations) increased from the first game to the third. Interestingly, the authors indicated that the playing time could be a key factor mediating the responses to congested periods. Another study by Doğramaci et al. [16] found a decline in sprint activity and an increase in walking distance across six games in four consecutive days of a tournament, thus confirming that the match demands were affected. The discrepancy between outcomes for Riberio et al. [13] and Doğramaci et al. [16] could be explained by the difference in the tournament schedule (i.e., three matches in four days versus six matches in four days), games played per day (i.e., one game per day versus up to three matches per day), and tracking technology system (inertial measurement units with ultrawideband tracking system technology versus video analysis). Lastly, Charlot et al. [15] observed an increase in RPE during a four-day Fédération Internationale de Football Association (FIFA) tournament, but a similar prevalence of high-intensity activities (same time spent >80% resting heart rate [HR]) and subjective well-being indices evaluated by the Hooper index across four matches. However, it is important to highlight that none of these studies [13,15,16] actually assessed players’ neuromuscular performance, physiological parameters and subjective responses during a congested period with specific tests such as the countermovement jump (CMJ), HR variability (HRV) or the Total Quality Recovery (TQR) scale. 

The testing of players during a congested period may be of interest for practitioners given that the cumulative futsal match loads could be detrimental for athletes’ performance and health [17]. In this regard, some of the abovementioned tests have been suggested as suitable options to track players’ response to fatigue and, consequently, minimize the risk of injury [18]. For instance, neuromuscular performance monitoring, through jump testing, is often used as a practical tool to periodically evaluate the response to training and competition stress, with a comprehensive analysis of kinetic variables during the jump being recommended to better detect neuromuscular impairments associated with acute or residual fatigue [19,20]. Regarding physiological parameters, HRV monitoring is a non-invasive, easy-to-implement procedure that takes into consideration the variation of time intervals (R-R intervals) between consecutive beats [21]; it has been shown to be a valid method for measuring cardiac parasympathetic–sympathetic nervous activity, and, consequently, the fatigue state of athletes [22,23]. Finally, the TQR scale is a questionnaire commonly used for measuring perceived recovery in sports [24], and studies [25,26] have demonstrated its validity for monitoring fatigue, as reductions in TQR scores have been found to be related to increases in creatine kinase, RPE and training load. 

To date, no studies have explored the effects of a congested competition period on the neuromuscular performance, physiological parameters and subjective recovery of young professional futsal players. Therefore, the main aims of this study were as follows: (1) to analyze the changes in CMJ jump-landing metrics, HRV and TQR scores, respectively, in a sample of under-19 (U19) male futsal players; and (2) to detect differences between those who played more minutes (HIGH_MIN_) and less minutes (LOW_MIN_) during the congested period. It was hypothesized that neuromuscular performance, physiological parameters and subjective recovery would be impaired during the four-day tournament period, and that the abovementioned variables would be significantly different between the two groups (HIGH_MIN_ and LOW_MIN_).

## 2. Materials and Methods

### 2.1. Study Design 

An observational study was designed to assess the neuromuscular performance (i.e., CMJ metrics), physiological parameters (i.e., HRV) and subjective recovery (i.e., TQR) of U19 male futsal players during the National Futsal Spanish Championship in April of 2022. The competitive calendar consisted of 3 matches in a 4-day period. During the Championship, players were tested on a daily basis with the same procedures carried out at the same hour, in the same order, and with the same researcher during the tournament (Figure 1). All evaluations (i.e., CMJ, HRV and TQR) were completed in the hotel where the team was hosted. It is important to highlight that 1 min HRV evaluation was selected because of the team´s logistic (just one person for 14 players, and balancing the players’ need to sleep versus the time required for data collection) and time restriction. All the evaluations took place in a high-stress-level environment in which many external aspects needed to be managed. Regarding the day 3 training session, players completed a recovery training session consisting of 10 min low-intensity running, 5 min dynamic stretching, 30 min footvolley on a tennis court, and then core and stretching exercises. 

### 2.2. Participants

Fourteen youth futsal players (age: 17.5 ± 0.5 years; body mass: 70.2 ± 8.5 kg; height: 1.80 ± 0.1 m) participated in the study. G*Power (version 3.1.9.7) (Heinrich-Heine Universität Düsseldorf—Düsseldorf, Germany) with the statistical test ANOVA, with repeated measures and within–between interaction, was used for the calculation of the sample size. A minimally interesting effect size (ES) (δ) 0.33, probability of error of 0.05, minimum desired power of 0.8, and a sample size of 14 futsal players were required. All athletes had ~8 years of experience in futsal and were involved in ~4 training sessions a week and one weekly match. Based on the playing time (registered manually by the strength and conditioning coach of the team), players were separated into two groups: those who played more minutes (HIGH_MIN_) (n = 7; body mass: 68.6 ± 8.8 kg; height: 1.81 ± 0.04 m) and less minutes (LOW_MIN_) (n = 7; body mass: 71.7 ± 7.5 kg; height: 1.77 ± 0.06 m). The groups were separated by median split and the threshold was defined as 43 min. The HIGH_MIN_ group had, on average, 65 ± 11 min effective playing time, and the LOW_MIN_ group played, on average, 22 ± 13 min; all of the players had playing time during the matches. The effective total playing time during the tournament was 120 min (i.e., 40 min per match). The study complied with ethical standards, was approved by an Institutional Research Ethics Committee with the registration number CE072008, and conformed to the recommendations of the Declaration of Helsinki. Players and legal guardians were made aware of the benefits and risks of this study and signed an informed consent form prior to participation. 

### 2.3. Procedures 

Vertical Jump Test: Athletes performed a CMJ test on a portable force platform (Kistler 9286BA, Kistler Group, Winterthur, Switzerland) following a standardized warm-up that consisted of dynamic stretching and lower-body activation exercises (3 sets of 12 reps of body-weight squats and lunges followed by skipping), followed by a test-specific warm-up (i.e., two sub-maximal attempts in all tested exercises), and each trial was separated by a 1 min rest interval. All data were exported and analyzed with a specific software (MARS, Kistler Group, Winterthur, Switzerland). Players were required to perform a downward movement followed by a complete, rapid extension of the lower limbs. The depth of the countermovement was self-selected to avoid changes in jumping coordination, and no other instructions were given regarding the landing technique. The hands were placed on the hips throughout the whole movement and players were directed to jump as high as possible and land close to the take-off point. They executed two maximal trials with a 1 min rest. The CMJ height was derived from an impulse–momentum equation [27], and the following additional CMJ variables were analyzed: relative peak power (PP_REL_), eccentric (Ecc) and concentric (Con) impulse, braking duration, and time to peak force. The mean data of the two jumps were used for analysis to reduce the error [28].

Heart Rate Variability: The resting HRV was obtained by the time elapsed between two successive R-waves of the QRS signal of the HR (R-R intervals) using an RS800cx (Polar Electro, Helsinki, Finland) HR monitor. The resting HRV was recorded every morning from 8:00 h to 8:20 h, in a dark room, as the players were woken up by the same strength and conditioning coach (Figure 1). During the R-R recordings, all players remained at rest for 1 min in the supine position following the standards set by a previous study [29], and a free breathing pattern was allowed during the recording. The correction of ectopic beats and erroneous signals was performed automatically using the manufacturer’s software (Kubios standard version 3.3.1 HRV Analysis, Finland) with a degree of correction <3% (medium threshold) for all recordings and an artifact acceptance threshold of 5%. Three data points were removed due to the sample artifact >5%. The resulting R-R intervals were examined in only a one-time domain index (i.e., root mean square difference of successive normal R-R intervals (RMSSD)). The RMSSD has been reported to reflect vagal modulation and to be related to training-induced effects.

Total Recovery Quality: All players answered the perception of recovery questionnaire (TQR) ranging from 6 (very very poorly recovered) to 20 (exceptionally well recovered) when they woke up and right before breakfast each day.

### 2.4. Statistical Analysis 

All data were analyzed using a statistical package (Jamovi, version 1.8, 2021). Linear mixed models were constructed to examine differences in CMJ kinetic variables according to the day and group, accounting for individual repeated measures. However, before running the linear mixed models, box plots and histograms were used to identify and exclude potentially influential data points. No outliers were detected in the analysis. Following this analysis, residual plots were visually inspected to determine deviations from homoscedasticity or normality. In all linear mixed models, group level (two levels) and the day (four levels) were used as the fixed effect in the model and the player as the random effect, with a random intercept and fixed slope. Pairwise comparisons were performed using the Bonferroni post hoc test. Hedge’s g effect sizes (ESs) with 95% confidence intervals (95% CI) were computed to determine the magnitude of every paired comparison and classified as follows: <0.2, trivial; 0.20–0.59, small; 0.60–1.19, moderate; 1.2–1.99, large and ≥2.0, very large [30]. The significance level was set at *p* ≤ 0.05.

## 3. Results

All tests used in this study displayed good levels of reliability (i.e., intraclass correlation coefficients (ICC) and coefficients of variation (CV)). Specifically, CMJ height presented 0.98 ICC, 3.6% CV, PP_REL_: 0.98 ICC, 2.8% CV, Ecc impulse: 0.96 ICC, 8.9% CV and Con impulse: 0.98 ICC, 3.2% CV, braking duration: 0.85 ICC, 8.7% CV, PF_Time_: 0.89 ICC, 11.4% CV, TQR: 0.83 ICC, 11.2% CV, RMSSD: 0.84 ICC, 25.1% CV, and LN RMSSD: 0.86 ICC and 5.4% CV.

Table 1 describes the heart rate variability, total recovery quality and countermovement-jump kinetic variables between groups and days. Figure 2 and Figure 3 present the CMJ PP_REL_ and TQR during the congested period in the HIGH_MIN_ and LOW_MIN_ groups, as well as the total sample (TOT_SAMPLE_).

Considering the TOT_SAMPLE_, a significant effect was found for day (*p* =< 0.001) on PP_REL_ and TQR. Specifically, PP_REL_ was significantly lower on Day 2 (*p* = 0.008; ES: 0.33; 95% CI: 0.16–1.36), Day 3 (*p* =< 0.001; ES: 0.36; 95% CI: 0.46–1.84) and Day 4 (*p* = 0.007; ES: 0.28; 95% CI: 0.45–1.90) when compared to Day 1. Furthermore, a significantly lower TQR score was found on Day 2 (*p* = 0.002; ES: 0.93; 95% CI: 0.16–1.50), Day 3 (*p* = 0.001; ES: 0.99; 95% CI: 0.11–1.35) and Day 4 (*p* = 0.013; ES: 0.85; 95% CI: 0.04–1.25) when compared to Day 1. Non-significant differences were found for the rest of the metrics.

The HIGH_MIN_ group presented significantly lower PP_REL_ on Day 3 (*p* = 0.004; ES: 0.62; 95% CI: 0.39–2.65) when compared to Day 1, and lower TRQ scores on Day 3 (*p* = 0.002; ES: 1.98; 95% CI: 0.18–2.46) and Day 4 (*p* = 0.003; ES: 2.25; 95% CI: 0.52–3.38) when compared to Day 1. Non-significant differences were found for the rest of the metrics and when considering the LOW_MIN_ group.

## 4. Discussion

The main objectives of the study were to examine the effects of a match-congested period on the neuromuscular performance (i.e., CMJ metrics), physiological parameters (i.e., HRV) and subjective recovery (i.e., TQR) of U19 male futsal players to identify differences between players with high and low playing time (i.e., HIGH_MIN_ and LOW_MIN_ groups, respectively) in terms of the abovementioned variables. The hypothesis was confirmed, as the findings revealed that a significant decrease was found in PP_REL_ on Days 2, 3, and 4 compared to Day 1, and in TQR scores on Days 2, 3, and 4 compared to Day 1 when analyzing the study sample as a whole (i.e., TOT_SAMPLE_). When considering the playing time, the HIGH_MIN_ group significantly reduced the mechanical power production (i.e., PP_REL_) on Day 3 when compared to Day 1, and presented lower perceived recovery (i.e., TQR) on Days 3 and 4 when compared to Day 1. In contrast, the LOW_MIN_ group did not present any significant change across all the metrics during the tournament, thus suggesting that the number of minutes played should be considered an important factor to take into account during match-congested periods in futsal. 

The results of this study showed that CMJ PP_REL_ was significantly impaired on Days 2, 3, and 4 compared to Day 1, considering the TOT_SAMPLE_, and on Day 3 (i.e., following two consecutive games on Days 1 and 2) only for the HIGH_MIN_ group compared to the first day. It is noteworthy to report that these declines in neuromuscular performance during the congested period seem to indicate that, as the competition load accumulated, players produced lower levels of force and did so more slowly. Nevertheless, CMJ height, Ecc and Con impulse, braking duration, and time to peak force had non-significant changes during the tournament in all groups (e.g., HIGH_MIN_, LOW_MIN_, and TOT_SAMPLE_). To the authors’ knowledge, no previous studies have evaluated CMJ metrics in young indoor soccer players during a congested period. However, in a study with elite basketball players, Doeven et al. [31] found no significant CMJ height differences between the congested and non-congested period, which partially supports the present findings. In line with this, Saidi et al. [32] also reported no changes in CMJ height after a congested schedule in elite soccer players. In contrast, Hernandez-Davo et al. [33] reported significant declines in CMJ height following a period of three matches per week compared with one match per week in young soccer players. Still, it is important to highlight that (1) the previous studies [31,32,33] only evaluated vertical jump ability before and after the congested period and not during the period itself (i.e., no daily measurements, as in the present research) and that (2) CMJ height might not be a good parameter to measure the players´ fatigue status [18,19]. According to the current study, congested periods induce acute neuromuscular fatigue in young futsal players with more on-court playing time, manifested as a decreased mechanical power output. From an applied perspective, CMJ may be appropriate to evaluate players´ neuromuscular performance and fatigue during congested periods, and potentially improve the team´s result, as has been found in a previous study with futsal players [34]. Furthermore, futsal coaches are advised to prioritize recovery strategies and closely monitor and share the playing time amongst players during periods with greater competition density.

Significant variations in HR are associated with overexertion [35] and symptoms of overtraining [36]. In the present study, HRV did not present significant changes throughout the tournament and no differences were observed between the groups. Similar findings were reported in previous studies [29,37,38,39]. For instance, Chen et al. [29] did not show variability in ultra-short-term (<1 min) and short-term (2–4 min) HRV measurements during a congested period in U20 national futsal players. Similarly, in Clemente et al. [38], non-significant changes were found in HRV across six training camps. Consistent with these findings, Ribeiro et al. [39] observed no significant changes in the HRV of ten young futsal players during three consecutive matches. The findings of the present study may be explained, at least in part, by the good training status of the athletes, the effective recovery strategies (i.e., active recovery, stretching/mobility, and cryotherapy post matches, nutrition and massage by the team´s physiotherapist) implemented, and the short duration of the congested period (i.e., 4 days). In an applied setting, HRV might not be the best option to detect significant negative alterations in the physiological system during a congested playing period, although the non-invasive and fast application of the measurement could encourage sports practitioners to use it for monitoring physiological changes. 

Regarding the TQR score, a worse perception of recovery was reported at the end of the tournament (Days 3 and 4) for the HIGH_MIN_ group and in the TOT_SAMPLE_ (Days 2, 3, and 4), which is consistent with previous studies [38,40,41,42]. However, the relationship between “well-being questionnaires” and variables such as delayed onset muscle soreness (DOMS) and RPE is equivocal. On the one hand, previous studies have reported an inverse correlation between well-being and these latter metrics [43,44]. On the other, Clemente et al. [14] found that DOMS of futsal players was higher during normal weeks when compared to weeks with two matches, and Doeven et al. [31] reported better-perceived well-being scores in basketball players during congested weeks. In light of these observations, it appears that the “type” of congested period may be a critical factor to consider since a congested period consisting of two matches per week is considerably different than a competitive scenario of three matches in four days. Therefore, based on the present data, it seems that a period of high match density negatively influences perceived recovery, mainly in players with greater match-playing time. Sports practitioners must prioritize and develop recovery strategies for this type of tournament, considering the athletes’ perception of recovery.

The present study is limited by its small sample size when comparing the two groups (i.e., HIGH_MIN_ and LOW_MIN_), highlighting that a potential underpower analysis and future research with more teams and players are warranted. Moreover, match external load metrics (e.g., total distance, high-speed running, sprints, accelerations, decelerations) were not considered, which could have helped to better understand the effects of consecutive matches on match performance; in addition, more thorough CMJ analysis with kinetic and kinematic variables may demonstrate differences when compared between the days and groups. Nevertheless, this is the first study to analyze the effects of congested match periods on neuromuscular and physiological parameters and recovery perception in young futsal players.

From an applied perspective, it is important for coaches to assess players’ neuromuscular performance (while considering metrics other than just jump height), as well as their perception of recovery during periods of match congestion. Strength and conditioning coaches and physiotherapists are advised to prioritize recovery methods to reduce the negative effects of back-to-back matches on players’ performance and well-being. Finally, the coaching staff is advised to pay special attention to players with the greatest exposure during matches in order to reduce the detrimental effects produced by the abovementioned variables.

## 5. Conclusions

In conclusion, congested periods (i.e., three matches in four days) elicit meaningful changes in neuromuscular and perceptual recovery in U19 futsal players. The main results of this study reveal that during a one-week tournament, CMJ PP_REL_ and TQR decreased significantly on Days 2, 3, and 4 when compared to Day 1, considering TOT_SAMPLE_. When accounting for the minutes played, the HIGH_MIN_ group showed a significant reduction in neuromuscular performance (i.e., PP_REL_) on Day 3, and in perceived recovery (i.e., TQR) on Days 3 and 4 when compared to Day 1. Nevertheless, HRV was not sensitive enough to detect negative alterations in physiological parameters in this type of tournament and context. 

## Figures and Tables

**Figure 1 sports-12-00139-f001:**
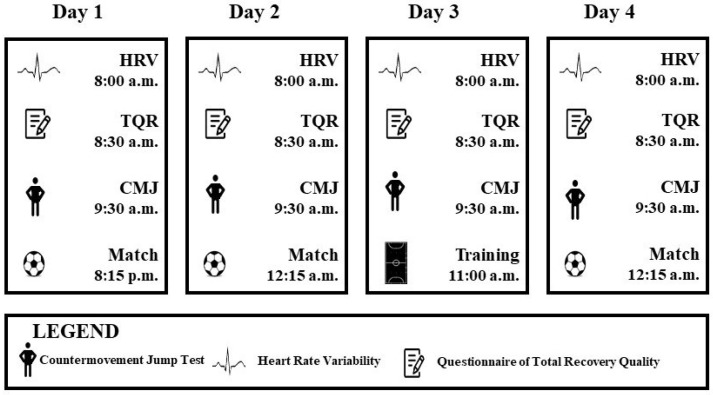
Study Design.

**Figure 2 sports-12-00139-f002:**
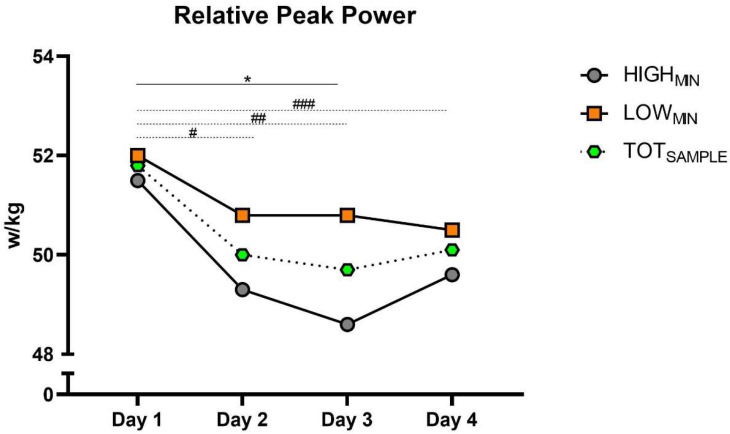
The groups (i.e., HIGH_MIN_, LOW_MIN_, and TOT_SAMPLE_) for PP_REL_ during the congested period. HIGH_MIN_: high minutes group; LOW_MIN_: low minutes group; PP_REL_: Relative peak power; TOT_SAMPLE_: Total sample. * Significant difference on Day 3 when compared to Day 1 for HIGH_MIN_ group. # Significant difference on Day 2 when compared to Day 1 for TOT_SAMPLE_. ## Significant difference on Day 3 when compared to Day 1 for TOT_SAMPLE_. ### Significant difference on Day 4 when compared to Day 1 for TOT_SAMPLE_.

**Figure 3 sports-12-00139-f003:**
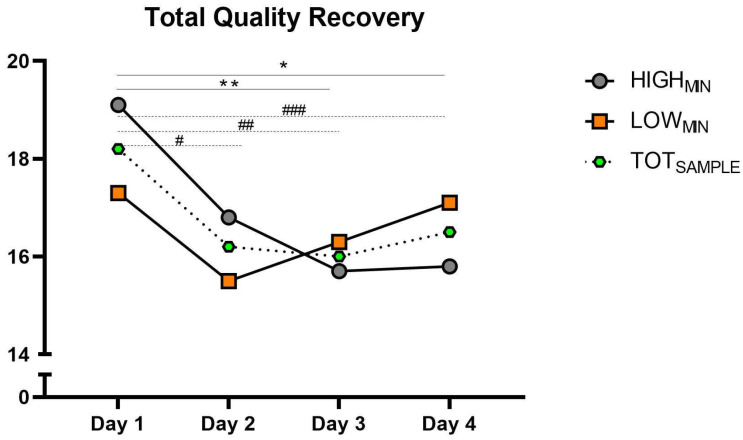
The groups (i.e., HIGH_MIN_, LOW_MIN_, and TOT_SAMPLE_) for TQR score during the congested period. HIGH_MIN_: high minutes group; LOW_MIN_: low minutes group; TQR: Total Quality Recovery; TOT_SAMPLE_: Total sample. * Significant difference on Day 3 when compared to Day 1 for HIGH_MIN_ group. ** Significant difference on Day 4 when compared to Day 1 for HIGH_MIN_ group. # Significant difference on Day 2 when compared to Day 1 for TOT_SAMPLE_. ## Significant difference on Day 3 when compared to Day 1 for TOT_SAMPLE_. ### Significant difference on Day 4 when compared to Day 1 for TOT_SAMPLE_.

**Table 1 sports-12-00139-t001:** Description of heart rate variability, total recovery quality and countermovement-jump kinetic variables between groups and days.

Variables	Units	Day 1		Day 2		Day 3		Day 4	
		HIGH_MIN_	LOW_MIN_	HIGH_MIN_	LOW_MIN_	HIGH_MIN_	LOW_MIN_	HIGH_MIN_	LOW_MIN_
RMSSD	ms	117 ± 34	123 ± 50	104 ± 53	118 ± 57	123 ± 61	113 ± 57	104 ± 48	113 ± 49
LN RMSSD	-	4.71 ± 0.38	4.73 ± 0.44	4.52 ± 0.54	4.66 ± 0.53	4.70 ± 0.55	4.62 ± 0.51	4.53 ± 0.53	4.63 ± 0.48
TQR	-	19.1 ± 1.0	17.3 ± 2.0	16.8 ± 2.0	15.5 ± 2.8	15.7 ± 2.2	16.3 ± 2.9	15.8 ± 1.8	17.1 ± 2.3
CMJ_HEIGHT_	cm	32.5 ± 5.2	33.6 ± 5.7	30.9 ± 5.3	33.1 ± 4.7	30.7 ± 5.9	33.8 ± 5.2	31.6 ± 4.7	33.5 ± 6.2
PP_REL_	W/kg	51.5 ± 4.4	52.0 ± 7.2	49.3 ± 4.6	50.8 ± 5.5	48.6 ± 4.9	50.8 ± 6.6	49.6 ± 5.6	50.5 ± 6.7
Ecc Impulse	Ns	−85.1 ± 18.2	−76.2 ± 23.1	−82.5 ± 13.3	−79.9 ± 26.6	−82.8 ± 15.3	−76.5 ± 30.3	−83.9 ± 14.7	−81.0 ± 26.9
Con Impulse	Ns	264 ± 39.5	253 ± 43.6	257 ± 36.0	255 ± 46.1	257 ± 38.6	252 ± 47.2	259 ± 37.1	257 ± 50.7
Braking duration	ms	277 ± 032	376 ± 064	299 ± 047	351 ± 038	304 ± 031	368 ± 066	292 ± 038	345 ± 048
PF_Time_	ms	551 ± 099	670 ± 144	598 ± 165	611 ± 117	591 ± 086	703 ± 157	587 ± 114	671 ± 164

CMJ: countermovement jump; Con: concentric; Ecc: eccentric; RMSSD: root mean square of successive RR interval differences; PF_TIME_: time to peak force; PP_REL_: relative peak power; TQR: total quality recovery.

## Data Availability

The datasets generated and/or analyzed during the present study are available from the corresponding author on request. The data are not publicly available due to privacy.

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
