# Peer review of "Does Total Playing Time Affect the Neuromuscular, Physiological, and Subjective Recovery of Futsal Players during a Congested Period?"

_sports, 2024, doi:10.3390/sports12050139_

Round 1
Reviewer 1 Report
Comments and Suggestions for Authors
General comments
This is a nice study that adds to the area, however there some areas for improvement around defining some of the measurements used and how you have calculated some of the measures. Essential it needs slightly more scientific exploration. Please see specific comments below.
Specific comments
Introduction
L35 – Could you add some velocities to the definitions of movement speed?
L36-37 – Could you add some frequency estimates for the movement types?
L39 – Check the SI units used here and on L43.
L47 – “This kind of fixtures..” doesn’t read appropriately, please amend.
L47 – I would delete “physical” to save confusion.
L48-49 – “worse match result” doesn’t read well, please amend.
L52 – What physical metrics did Ribeiro observe?
L53-54 – So, there was no fatigue if they increased? Maybe pacing?
L56 – Change the word “research” for “study”.
L52-58 – Add some explanation to why the differences between the studies highlighted?
L61 – What is the Hooper index?
L73 – I would remove the word “land” with regards to the CMJ, as you have not observed any landing metrics.
Methods
L91 – It is an observational study design.
L94 – For clarity, it wasn’t 3 matches on 4 consecutive days as there is a break. Change to something like “3 matches in a 4 day period”
L94 – I would also like to see some mention of the training on day 3, what was it how intense etc?
L106 – What literature did you use to determine the minimally-interesting effect size of 0.33?
L109-113 – It is great you have a-priori sample size estimation for the anova, but do you have sufficient statistical power for the between High and Low differences?
L133 – Using the cue to “jump as high as possible” could result in strategy changes (i.e. more compliant), why did you decide on this cue?
L134 – Add all reliability measures to the results.
L136 – How did you define the concentric and eccentric phases? This is key within the literature as there are various definitions which are not identical. Moreover, these terms should be avoided as we do not know the true muscle action.
McMahon, John & Suchomel, Timothy & Lake, Jason & Comfort, Paul. (2018). Understanding the Key Phases of the Countermovement Jump Force-Time Curve. Strength and conditioning journal. 40. 10.1519/SSC.0000000000000375.
Hahn, D. (2023), On the phase definitions of counter movement jumps. Scand J Med Sci Sports, 33: 359-360. https://doi.org/10.1111/sms.14288
L136-137 – “force impulse” does not make sense as a term, as impulse is force x time.
L137 – Check the punctuation with the capital T
L161 – Were any outliers excluded?
L165 – What post-hoc tests were ran?
L165 – Hedge’s g effect size would be more appropriate as it accounts for sample size, especially smaller samples.
Results
Table 1 – Why choose relative peak power as a metric, as it has minimal implication on jump performance? Moreover, you have used a cue that encourages a compliant, slow strategy this needs to be explored.
Figure 2 and Figure 3 – Remove the large heading for the figures.
Discussion
L213 – Remove landing.
L215 – A study is inanimate; it cannot aim. Please amend accordingly.
L230 – How do you know if it was lower levels of force slower? You haven’t observed the metrics independently.
L236 – Any differences in how the CMJs were assessed?
L244 – Why jump height? Please see;
Spencer, R.; Sindall, P.; Hammond, K.M.; Atkins, S.J.; Quinn, M.; McMahon, J.J. Changes in Body Mass and Movement Strategy Maintain Jump Height Immediately after Soccer Match. Appl. Sci. 2023, 13, 7188. https://doi.org/10.3390/app13127188
L257 – Why use a short 1 min test period then?
L263 – You should describe the exact recovery modalities used, this is crucial.
L267-268 – This sentence seems incomplete.
L275-277 – Why did these studies find this? Was it potentially less focused on the feel more focused on the performance? Or was there an enhanced focus on recovery in congested periods?
L285 – I would be specific here, you have the required sample based of you’re a-priori observations for the anova, what you are limited on is the paired differences so I would specifically highlight this and mention potential for underpowered.
L293 – It is probably essential to observe other metrics. See the paper suggested above.
L303 – Elsewhere PPREL the REL has been subscript.
L307 – Change the word “sensible” for “sensitive”
Comments on the Quality of English LanguageGenerally it is ok, there are a few areas that need improvement just to aid in the readers understanding.
Author Response
Reviewer 1.
General comments
This is a nice study that adds to the area, however there some areas for improvement around defining some of the measurements used and how you have calculated some of the measures. Essential it needs slightly more scientific exploration. Please see specific comments below.
Authors Response: Dear Reviewer, thank you for the thorough review of the manuscript. We really appreciate your suggestions, definitely increased the quality of the manuscript, and all of them were addressed accordingly.
Thanks in advance.
Specific comments
Introduction
L35 – Could you add some velocities to the definitions of movement speed?
Authors Response: All the velocities have been added and now it stated: “Futsal is characterized as a high-intensity intermittent sport, with high physical match-play demands, that involve the aerobic-anaerobic (e.g., walking [0-6 km·h-1], low- [6.1-12 km·h-1], medium- [12.1-18 km·h-1], and high-intensity running [>18 km·h-1]) …”
L36-37 – Could you add some frequency estimates for the movement types?
Authors Response: All the frequencies have been added and not it stated: “and neuromuscular (e.g., accelerations [>2 m·s-2], decelerations [> -2 m·s-2], repetitive high-intensity sprints, and changes of direction) systems …”
L39 – Check the SI units used here and on L43.
Authors Response: Thanks for the detection, the change has been made.
L47 – “This kind of fixtures..” doesn’t read appropriately, please amend.
Authors Response: We agree with the reviewer´s suggestion and now it is stated as the following: “These congested periods …”
L47 – I would delete “physical” to save confusion.
Authors Response: The suggestion has been made.
L48-49 – “worse match result” doesn’t read well, please amend.
Authors Response: We agree and now it is stated as “match result”.
L52 – What physical metrics did Ribeiro observe?
Authors Response: Authors added the physical metrics and now it is stated as following: “In a study with elite players, Ribeiro et al. [12] found that congested fixtures did not negatively affect physical match performance (i.e., total distance covered, high-speed running, sprinting, accelerations and decelerations), …”.
L53-54 – So, there was no fatigue if they increased? Maybe pacing?
Authors Response: Exactly, it could be explained by pacing and the effective playing time as the authors indicate, and it is written in the manuscript as well (i.e., the authors indicated that playing time could be a key factor mediating the responses to congested periods.).
L56 – Change the word “research” for “study”.
Authors Response: The change has been made.
L52-58 – Add some explanation to why the differences between the studies highlighted?
Authors Response: We agree with reviewer´s comment and now it is stated as the following: “These discrepancies between the two studies [12, 15] could be explained by the difference of the tournaments´ schedule (i.e., 3 matches in 4 days versus 6 matches in 4 days), games per day (i.e, one game per day versus up to three matches per day), and tracking technology system (inertial measurement units with ultrawideband tracking system technology versus video analysis).”.
Thanks in advance.
L61 – What is the Hooper index?
Authors Response: The reviewer is right and more information have been amended and now it is stated as “subjective well-being indices evaluated by Hooper index”
L73 – I would remove the word “land” with regards to the CMJ, as you have not observed any landing metrics.
Authors Response: We agree and the word has been removed throughout the manuscript. Thanks.
Methods
L91 – It is an observational study design.
Authors Response: The reviewer is right, and the change has been made.
L94 – For clarity, it wasn’t 3 matches on 4 consecutive days as there is a break. Change to something like “3 matches in a 4 day period”
Authors Response: Great idea! We completely agree and the change has been made. Thanks
L94 – I would also like to see some mention of the training on day 3, what was it how intense etc?
Authors Response: We completely agree with the reviewer´s suggestion and it is important information that should be given to the readers. We added all the information about the training in the 3rd Day and now it is stated as the following: “Regarding the day 3 training session, players realized a recovery training session, consisted of 10 min low-intensity running, 5 min dynamic stretching, 30 min foot-volley in a tennis court, and then core and stretching exercises.”
L106 – What literature did you use to determine the minimally-interesting effect size of 0.33?
Authors Response: Thanks for the opportunity to clarify this. The effect size of 0.33 for the power analysis indicated that a minimum sample size of 14 participants in total would be needed, we have not based on a previous literature. We are open to further change or suggestion. Thanks in advance.
L109-113 – It is great you have a-priori sample size estimation for the anova, but do you have sufficient statistical power for the between High and Low differences?
Authors Response: Unfortunately, we did not perform G*Power calculation for the two groups because this analysis came up from the sample that we already had. We acknowledge that one of the limitations of the study is the sample size. As stated before, we are really open to further change or reviewer’s suggestions. Thanks!
L133 – Using the cue to “jump as high as possible” could result in strategy changes (i.e. more compliant), why did you decide on this cue?
Authors Response: We have used this verbally cue mainly based on this study “Krzyszkowski, J., Chowning, L. D., & Harry, J. R. (2022). Phase-Specific Verbal cue effects on countermovement jump performance. The Journal of Strength & Conditioning Research, 36(12), 3352-3358.”, and as well many other studies in futsal have been used this cue. Thanks for the opportunity to clarify this.
L134 – Add all reliability measures to the results.
Authors Response: The authors prefer to keep the reliability measures in the methods section as they are not the main objectives and results of the study, and keep the results section focus on the main findings. However, we are open to any suggestion regarding this point. Thanks for the understanding.
L136 – How did you define the concentric and eccentric phases? This is key within the literature as there are various definitions which are not identical. Moreover, these terms should be avoided as we do not know the true muscle action.
McMahon, John & Suchomel, Timothy & Lake, Jason & Comfort, Paul. (2018). Understanding the Key Phases of the Countermovement Jump Force-Time Curve. Strength and conditioning journal. 40. 10.1519/SSC.0000000000000375.
Hahn, D. (2023), On the phase definitions of counter movement jumps. Scand J Med Sci Sports, 33: 359-360. https://doi.org/10.1111/sms.14288
Authors Response: We totally agree with the reviewer and thanks for the opportunity to clarify this issue. The eccentric phase is calculated by negative velocity starting from the point where a 20 N threshold is exceeded until velocity = 0 m·s−1 and the concentric phase from positive velocity from = m·s−1 until takeoff.
L136-137 – “force impulse” does not make sense as a term, as impulse is force x time.
Authors Response: We totally agree with reviewer’s suggestion and we changed as “impulse” throughout the manuscript. Thanks for the suggestion.
L137 – Check the punctuation with the capital T
Authors Response: We are sorry for this typo mistake; we have corrected it.
L161 – Were any outliers excluded?
Authors Response: Thanks for the opportunity to add this in the manuscript. As we have amended to manuscript, and no outliers were detected in the analysis.
L165 – What post-hoc tests were ran?
Authors Response: Thanks for the opportunity to clarify this. We used a Bonferroni post-hoc analysis, and now is stated in the manuscript as the following “Pairwise comparisons were performed using Bonferroni post-hoc test.”.
L165 – Hedge’s g effect size would be more appropriate as it accounts for sample size, especially smaller samples.
Authors Response: We really appreciate for this suggestion. We agree with the reviewer’s suggestion and change of the calculation of the effect size by Hedge’s g has been applied and changed in the manuscript. Thanks!
Results
Table 1 – Why choose relative peak power as a metric, as it has minimal implication on jump performance? Moreover, you have used a cue that encourages a compliant, slow strategy this needs to be explored.
Authors Response: The authors decided to utilize the relative peak power because they believe the power could be an important indicator to jump according to the reference “Bishop, C., Jordan, M., Torres-Ronda, L., Loturco, I., Harry, J., Virgile, A., ... & Comfort, P. (2023). Selecting metrics that matter: comparing the use of the countermovement jump for performance profiling, neuromuscular fatigue monitoring, and injury rehabilitation testing. Strength & Conditioning Journal, 45(5), 545-553.” and in the futsal specifically.
Figure 2 and Figure 3 – Remove the large heading for the figures.
Authors Response: Authors prefer to maintain the headings in the figures to be clear for the reader which variable is for each figure. Thanks for the understanding.
Discussion
L213 – Remove landing.
Authors Response: It is removed from the previous comment.
L215 – A study is inanimate; it cannot aim. Please amend accordingly.
Authors Response: We agree with reviewer´s suggestions and we changed it accordingly and now it is stated as the following: The main objectives of study were to examine the effects of a match-congested period on neuromuscular performance (i.e., CMJ metrics), physiological parameters (i.e., HRV) and subjective recovery (i.e., TQR) of U19 male futsal players to identify differ-ences between players with high and low playing time (i.e., HIGHMIN and LOWMIN groups, respectively) on the abovementioned variables.
L230 – How do you know if it was lower levels of force slower? You haven’t observed the metrics independently.
Authors Response: Thanks for the opportunity to clarify this issue. We totally agree and it was written because of the decrement in relative peak power. We are open to adjust it.
L236 – Any differences in how the CMJs were assessed?
Authors Response: No difference how the CMJ was assessed.
L244 – Why jump height? Please see;
Spencer, R.; Sindall, P.; Hammond, K.M.; Atkins, S.J.; Quinn, M.; McMahon, J.J. Changes in Body Mass and Movement Strategy Maintain Jump Height Immediately after Soccer Match. Appl. Sci. 2023, 13, 7188. https://doi.org/10.3390/app13127188
Authors Response: We highlight exactly what the abovementioned study indicated, that only CMJ height is not sensitive enough to detect fatigue because different strategies could be used to reach the same height.
L257 – Why use a short 1 min test period then?
Authors Response: Thanks for the opportunity to clarify this. We selected to measure 1 min because of logistic (just one person for 14 players, and trying players do not lose a lot of sleeping time) and time restriction. All the evaluations took place in high level environment when we could have managed many external aspects. Thanks.
L263 – You should describe the exact recovery modalities used, this is crucial.
Authors Response: We agree that it is very important. All the recovery modalities are mentioned in the manuscript (i.e., active recovery, stretching/mobility, and cryotherapy post matches, nutrition and massage by the team´s physiotherapist), and it is true that is crucial and for this reason are mentioned in the manuscript.
L267-268 – This sentence seems incomplete.
Authors Response: We agree with reviewer’s suggestion and now it is stated as “In an applied setting, HRV might not be the best option to detect significant negative alterations in physiological system during congested period, although the non-invasive and fast application of the measurement could make sports practitioners to use it for monitoring physiological changes.”
L275-277 – Why did these studies find this? Was it potentially less focused on the feel more focused on the performance? Or was there an enhanced focus on recovery in congested periods?
Authors Response: Exactly, authors as well believe that these studies found this because during the congested period coaches focus on recovery and tactical preparation for the next match instead of increasing the physical capacities and / or physical preparation.
L285 – I would be specific here, you have the required sample based of you’re a-priori observations for the anova, what you are limited on is the paired differences so I would specifically highlight this and mention potential for underpowered.
Authors Response: The reviewer is 100% right and we have changed according to his suggestion. Now, it stated as: “The present study is limited by its small sample size when comparing between the two groups (i.e., HIGHMIN and LOWMIN) highlighting a potential underpower analysis”
L293 – It is probably essential to observe other metrics. See the paper suggested above.
Authors Response: We totally agree with the reviewer, but unfortunately, with the software that it has been used, are given only those that we used.
L303 – Elsewhere PPREL the REL has been subscript.
Authors Response: We are sorry for this mistake. We have corrected throughout the manuscript.
L307 – Change the word “sensible” for “sensitive”
Authors Response: Correct, we changed it.
Reviewer 2 Report
Comments and Suggestions for Authors
Dear Authors,
Please see the attached comments. Thank you.
General Comments
The article has a very current research topic in football codes, specifically the effect of congested schedules on neuromuscular, physiological, and subjective variables. Furthermore, the variables include assessment methods (i.e., CMJ) and monitoring methods (HRV, TQR). The independent variable also subdivides the groups into more minutes (HIGHMIN) and less minutes (LOWMIN), something very interesting with quite research-gap practice and a good orientation for research aims: analyse the 85 changes in CMJ jump-landing metrics, HRV and TQR scores, respectively, in a sample of 86 under-19 (U19) male futsal players; and 2) detect differences between those who played 87 more minutes (HIGHMIN) and less minutes (LOWMIN) during the congested period.
Specific comments
Introduction:
· The introduction is strong and interesting. I just recommend adding some inferences about previous studies analysing player starting status in futsal (please, see: http://dx.doi.org/10.7575/aiac.ijkss.v.10n.2p.42).
· Specifically, this topic has been studied a lot in football with fixed substitutions, but in futsal the study is still almost non-existent. That's your research gap, but you should frame previous studies (please, see: https://doi.org/10.3390/ijerph191811611).
· Please, add the research hypothesis after the research aims.
Methods:
· The first subchapter should be "2.1 Participants" and only then should the study design be described (and the procedures described in the same subchapter).
· Some important points in the study design remain to be described: how many minutes of the game were played on average by each HIGHMIN and LoW player? Did they all take part in the same games? If the substitutions were rolling, how many times did they come on and off (this may have an impact on their ability to run repeated sprints). What is the level between the teams, match outcome, classification? Contextual variables have a high impact on physical performance.
· The cut-off values for interpreting the results of the vertical jump test, HRV and TQR have yet to be added.
· The statistical procedures are strong. It would have been interesting to add a bootstrap since we want to monitor a congested schedule, however we only evaluate individuals over 4 days.
Results:
· The results are interesting, extensive and well presented.
Discussion and conclusions:
· In the discussion, the first paragraph should confirm or reject the null hypothesis. The relationship between your study and others is interesting and in-depth. However, the results lack a practical application. What is the optimum value for each player to play? Which bout periods should they play? How does neuromuscular and physiological performance and the perception of recovery change over the course of the game? Or more specifically, the magnitude of the differences in the variables in High or Low min, what will it represent in the management of the player's playing minutes during the game and later during training? I therefore recommend that you only consider expanding future prospects and practical applications (please, see: https://doi.org/10.1186/s40064‑016‑2327‑x | https://doi.org/10.1016/j.physbeh.2019.01.001
References:
· The references should be improved
Author Response
Reviewer 2.
Dear Authors,
Please see the attached comments. Thank you.
General Comments
The article has a very current research topic in football codes, specifically the effect of congested schedules on neuromuscular, physiological, and subjective variables. Furthermore, the variables include assessment methods (i.e., CMJ) and monitoring methods (HRV, TQR). The independent variable also subdivides the groups into more minutes (HIGHMIN) and less minutes (LOWMIN), something very interesting with quite research-gap practice and a good orientation for research aims: analyse the 85 changes in CMJ jump-landing metrics, HRV and TQR scores, respectively, in a sample of 86 under-19 (U19) male futsal players; and 2) detect differences between those who played 87 more minutes (HIGHMIN) and less minutes (LOWMIN) during the congested period.
Authors Response: The authors are very grateful for reviewer’s suggestions and comments, that have significantly enhanced its quality, and we have addressed each of them. Thank you in an advance.
Specific comments
Introduction:
- The introduction is strong and interesting. I just recommend adding some inferences about previous studies analysing player starting status in futsal (please, see: http://dx.doi.org/10.7575/aiac.ijkss.v.10n.2p.42).
Authors Response: Dear reviewer, thanks for your comment and the suggestion. Could you please specify where to cite this reference during the introduction because it presents important information about intervention program during the pre-season and in-season. We are really sorry and authors are open to any other adjustments.
- Specifically, this topic has been studied a lot in football with fixed substitutions, but in futsal the study is still almost non-existent. That's your research gap, but you should frame previous studies (please, see: https://doi.org/10.3390/ijerph191811611).
Authors Response: Thank you very much for the suggesting reference. It has been added.
- Please, add the research hypothesis after the research aims.
Authors Response: Thank you for the comment. The hypothesis has been added and now it stated as “It was hypothesized that neuromuscular performance, physiological parameters and subjective recovery would be impaired during the 4-day period, and it would be sig-nificant difference on the abovementioned variables between the two groups (HIGHMIN and LOWMIN).”
Methods:
- The first subchapter should be "2.1 Participants" and only then should the study design be described (and the procedures described in the same subchapter).
Authors Response: The authors prefer to maintain first the study design and then participants as it is recommended from the journal. Furthermore, the procedures should be in different section and with the study design in order to be clear to the reader. We are really sorry but we are open for any further change.
- Some important points in the study design remain to be described: how many minutes of the game were played on average by each HIGHMIN and LoW player? Did they all take part in the same games? If the substitutions were rolling, how many times did they come on and off (this may have an impact on their ability to run repeated sprints). What is the level between the teams, match outcome, classification? Contextual variables have a high impact on physical performance.
Authors Response: Thank you very much for the opportunity to clarify this issue.
Firstly, all the information about average playing time is given in the manuscript.
Regarding whether players took part in the same games, we totally agree and it is important information and has been added to the manuscript.
Authors do not have the information how many times players rolling in and out of the court. Moreover, it is written in the limitations section, we didn´t monitor the external load that 100% agree that was influenced.
Lastly, all the teams had a high level as it was for National Futsal Spanish Championship.
Thanks in advance.
- The cut-off values for interpreting the results of the vertical jump test, HRV and TQR have yet to be added.
Authors Response: We are sorry but we don´t understand this suggestion. Why authors should use the cut-off values? The cut-off values used for the two groups those who played more and less minutes. Thanks, and we are open to further discussion.
- The statistical procedures are strong. It would have been interesting to add a bootstrap since we want to monitor a congested schedule, however we only evaluate individuals over 4 days.
Authors Response: Thank you very much for the opportunity to clarify this. A bootstrap analysis has been used before and no differences were observed in the results because using Linear Mixed Model there is a high number of data points and there is effect of parametric or non-parametric tests.
Results:
- The results are interesting, extensive and well presented.
Authors Response: Thanks for your kind words.
Discussion and conclusions:
- In the discussion, the first paragraph should confirm or reject the null hypothesis. The relationship between your study and others is interesting and in-depth. However, the results lack a practical application. What is the optimum value for each player to play? Which bout periods should they play? How does neuromuscular and physiological performance and the perception of recovery change over the course of the game? Or more specifically, the magnitude of the differences in the variables in High or Low min, what will it represent in the management of the player's playing minutes during the game and later during training? I therefore recommend that you only consider expanding future prospects and practical applications (please, see: https://doi.org/10.1186/s40064‑016‑2327‑x | https://doi.org/10.1016/j.physbeh.2019.01.001
Authors Response:
We totally agree with the reviewer’s suggestion and the confirm of the null hypothesis has been added in the manuscript.
Regarding the practical applications are presented throughout the manuscript, please check lines 238-240, 265-267, 280-284, 298-300 and a whole paragraph 309-316. Moreover, the authors prefer do not mention a specific or optimum minutes of playing time that player should play or bout of play because we cannot specify it. We have considered the reference “Clemente FM, Martinho R, Calvete F, Mendes B. Training load and well-being status variations of elite futsal players across a full season: Comparisons between normal and congested weeks. Physiol Behav. 2019 Mar;201:123–9” that the reviewer suggest.
Thanks in advance.
References:
- The references should be improved
Authors Response: Thanks for your comment, and we think that following the author’s suggested references, the section has been improved.
Round 2
Reviewer 1 Report
Comments and Suggestions for Authors
General comments
I greatly appreciate the effort the authors have gone to at making the suggested changes and edits and the manuscript is substantially improved. I do have a few minor comments suggestions and some of these are based of previously stated comments. But overall, what has been added and changed has substantially improved the manuscript.
Specific comments
L119 – What literature did you use to determine the minimally-interesting effect size of 0.33?
Authors Response: Thanks for the opportunity to clarify this. The effect size of 0.33 for the power analysis indicated that a minimum sample size of 14 participants in total would be needed, we have not based on a previous literature. We are open to further change or suggestion. Thanks in advance.
I appreciate the clarity, but if you have not used an effect size from the literature you have just randomly selected a number to attain a minimum sample size required. You need to look at what has been observed previously within the literature.
E.g. the below study found an effect of 0.22, so that. would be used in the a-prori sample size estimation.
Franceschi A, Robinson MA, Owens D, Brownlee T, Ferrari Bravo D and Enright K (2023) Reliability and sensitivity to change of post-match physical performance measures in elite youth soccer players. Front. Sports Act. Living 5:1173621. doi: 10.3389/fspor.2023.1173621
Ideally you would look for a meta-analysis on this subject but that can be difficult.
Add all reliability measures to the results.
Authors Response: The authors prefer to keep the reliability measures in the methods section as they are not the main objectives and results of the study, and keep the results section focus on the main findings. However, we are open to any suggestion regarding this point. Thanks for the understanding.
I understand why you maybe reluctant to add the reliability to the results section, however you can use a more robust method of interpretation within the results that you cannot do here.
Koo TK, Li MY. A Guideline of Selecting and Reporting Intraclass Correlation Coefficients for Reliability Research. J Chiropr Med. 2016 Jun;15(2):155-63. doi: 10.1016/j.jcm.2016.02.012. Epub 2016 Mar 31. Erratum in: J Chiropr Med. 2017 Dec;16(4):346. PMID: 27330520; PMCID: PMC4913118.
L136 – How did you define the concentric and eccentric phases? This is key within the literature as there are various definitions which are not identical. Moreover, these terms should be avoided as we do not know the true muscle action.
McMahon, John & Suchomel, Timothy & Lake, Jason & Comfort, Paul. (2018). Understanding the Key Phases of the Countermovement Jump Force-Time Curve. Strength and conditioning journal. 40. 10.1519/SSC.0000000000000375.
Hahn, D. (2023), On the phase definitions of counter movement jumps. Scand J Med Sci Sports, 33: 359-360. https://doi.org/10.1111/sms.14288
Authors Response: We totally agree with the reviewer and thanks for the opportunity to clarify this issue. The eccentric phase is calculated by negative velocity starting from the point where a 20 N threshold is exceeded until velocity = 0 m·s−1 and the concentric phase from positive velocity from = m·s−1 until takeoff.
I appreciate the further explanation here, however this means the “eccentric phase” includes an unweighting phase i.e. where there is no force production just the body relaxing. I would also see the work by Hahn on the naming.
Discussion
L244 – Why jump height? Please see;
Spencer, R.; Sindall, P.; Hammond, K.M.; Atkins, S.J.; Quinn, M.; McMahon, J.J. Changes in Body Mass and Movement Strategy Maintain Jump Height Immediately after Soccer Match. Appl. Sci. 2023, 13, 7188. https://doi.org/10.3390/app13127188
Authors Response: We highlight exactly what the abovementioned study indicated, that only CMJ height is not sensitive enough to detect fatigue because different strategies could be used to reach the same height.
That is great, but why not look at the different strategies involved? Or any changes in body mass?
L257 – Why use a short 1 min test period then?
Authors Response: Thanks for the opportunity to clarify this. We selected to measure 1 min because of logistic (just one person for 14 players, and trying players do not lose a lot of sleeping time) and time restriction. All the evaluations took place in high level environment when we could have managed many external aspects. Thanks.
Thanks for the clarification, I would add this to the manuscript it is great information.
L293 – It is probably essential to observe other metrics. See the paper suggested above.
Authors Response: We totally agree with the reviewer, but unfortunately, with the software that it has been used, are given only those that we used.
I would add this to the limitation, I would also look into the software to see if you can export the raw data out and manually analyse as you are leaving a lot on the table that would be of interest.
Author Response
General comments
I greatly appreciate the effort the authors have gone to at making the suggested changes and edits and the manuscript is substantially improved. I do have a few minor comments suggestions and some of these are based of previously stated comments. But overall, what has been added and changed has substantially improved the manuscript.
Authors Response: The authors really appreciate the reviewer’s time and effort to review the manuscript and all the suggestions and comments addressed below.
Thanks in advance for the contribution on this process.
Specific comments
L119 – What literature did you use to determine the minimally-interesting effect size of 0.33?
Authors Response: Thanks for the opportunity to clarify this. The effect size of 0.33 for the power analysis indicated that a minimum sample size of 14 participants in total would be needed, we have not based on a previous literature. We are open to further change or suggestion. Thanks in advance.
I appreciate the clarity, but if you have not used an effect size from the literature you have just randomly selected a number to attain a minimum sample size required. You need to look at what has been observed previously within the literature.
E.g. the below study found an effect of 0.22, so that. would be used in the a-prori sample size estimation.
Franceschi A, Robinson MA, Owens D, Brownlee T, Ferrari Bravo D and Enright K (2023) Reliability and sensitivity to change of post-match physical performance measures in elite youth soccer players. Front. Sports Act. Living 5:1173621. doi: 10.3389/fspor.2023.1173621
Ideally you would look for a meta-analysis on this subject but that can be difficult.
Authors Response: Dear reviewer, thank you for the clarification. We will proceed this way for future occasions. However, we used the effect size of 0.33 based on our own “in-house” data (unpublished) and also because it falls within the range determined for small changes in highly trained athletes (0.25-0.50) according to classification by Rhea “Rhea, M. R. (2004). Determining the magnitude of treatment effects in strength training research through the use of the effect size. The Journal of Strength & Conditioning Research, 18(4), 918-920.”.
Add all reliability measures to the results.
Authors Response: The authors prefer to keep the reliability measures in the methods section as they are not the main objectives and results of the study, and keep the results section focus on the main findings. However, we are open to any suggestion regarding this point. Thanks for the understanding.
I understand why you maybe reluctant to add the reliability to the results section, however you can use a more robust method of interpretation within the results that you cannot do here.
Koo TK, Li MY. A Guideline of Selecting and Reporting Intraclass Correlation Coefficients for Reliability Research. J Chiropr Med. 2016 Jun;15(2):155-63. doi: 10.1016/j.jcm.2016.02.012. Epub 2016 Mar 31. Erratum in: J Chiropr Med. 2017 Dec;16(4):346. PMID: 27330520; PMCID: PMC4913118.
Authors Response:We agree with the reviewer´s suggestion and the ICC and CV have been amended in the results, and it is stated as the following: “All tests used in this study displayed good levels of reliability (i.e., intraclass correlation coefficients [ICC] and coefficients of variation [CV]). Specifically, CMJ height presented 0.98 ICC, 3.6% CV, PPREL: 0.98 ICC, 2.8% CV, Ecc impulse: 0.96 ICC, 8.9% CV and Con impulse: 0.98 ICC,3.2% CV, braking duration: 0.85 ICC, 8.7% CV, PFTime: 0.89 ICC, 11.4% CV, TQR: 0.83 ICC, 11.2% CV, RMSSD: 0.84 ICC, 25.1% CV, and LN RMSSD: 0.86 ICC and 5.4% CV.”
L136 – How did you define the concentric and eccentric phases? This is key within the literature as there are various definitions which are not identical. Moreover, these terms should be avoided as we do not know the true muscle action.
McMahon, John & Suchomel, Timothy & Lake, Jason & Comfort, Paul. (2018). Understanding the Key Phases of the Countermovement Jump Force-Time Curve. Strength and conditioning journal. 40. 10.1519/SSC.0000000000000375.
Hahn, D. (2023), On the phase definitions of counter movement jumps. Scand J Med Sci Sports, 33: 359-360. https://doi.org/10.1111/sms.14288
Authors Response: We totally agree with the reviewer and thanks for the opportunity to clarify this issue. The eccentric phase is calculated by negative velocity starting from the point where a 20 N threshold is exceeded until velocity = 0 m·s−1 and the concentric phase from positive velocity from = m·s−1 until takeoff.
I appreciate the further explanation here, however this means the “eccentric phase” includes an unweighting phase i.e. where there is no force production just the body relaxing. I would also see the work by Hahn on the naming.
Authors Response:Dear reviewer, exactly, it includes the unweighting phase. Thanks in advance.
Discussion
L244 – Why jump height? Please see;
Spencer, R.; Sindall, P.; Hammond, K.M.; Atkins, S.J.; Quinn, M.; McMahon, J.J. Changes in Body Mass and Movement Strategy Maintain Jump Height Immediately after Soccer Match. Appl. Sci. 2023, 13, 7188. https://doi.org/10.3390/app13127188
Authors Response: We highlight exactly what the abovementioned study indicated, that only CMJ height is not sensitive enough to detect fatigue because different strategies could be used to reach the same height.
That is great, but why not look at the different strategies involved? Or any changes in body mass?
Authors Response: Thanks for the opportunity clarify this issue. Firstly, because of software limitation authors did not select more variables closing to the strategy as a center of mass displacement, that as McMahon, and other authors demonstrated important variable. Although, acknowledge the limitation of the current variables, we tried to identify fatigue by Ecc and Con impulse, Braking duration and time to peak force according to Bishop, C., Jordan, M., Torres-Ronda, L., Loturco, I., Harry, J., Virgile, A., ... & Comfort, P. (2023). Selecting metrics that matter: comparing the use of the countermovement jump for performance profiling, neuromuscular fatigue monitoring, and injury rehabilitation testing. Strength & Conditioning Journal, 45(5), 545-553. Regarding the body mass would be a good idea if it would have been evaluated acute post-game evaluations compared to pre -game instead of daily basis following the breakfast as we designed it (figure 1). Thanks in advance.
L257 – Why use a short 1 min test period then?
Authors Response: Thanks for the opportunity to clarify this. We selected to measure 1 min because of logistic (just one person for 14 players, and trying players do not lose a lot of sleeping time) and time restriction. All the evaluations took place in high level environment when we could have managed many external aspects. Thanks.
Thanks for the clarification, I would add this to the manuscript it is great information.
Authors Response: Great idea, we agree with reviewer´s suggestion. The phrase has been added and now it is stated as the following: “It is important to highlight, that 1 min HRV evaluation was selected because of team´s logistic (just one person for 14 players, and trying players do not lose a lot of sleeping time) and time restriction. All the evaluations took place in high level environment when many external aspects should have been managed.” Thanks in advance.
L293 – It is probably essential to observe other metrics. See the paper suggested above.
Authors Response: We totally agree with the reviewer, but unfortunately, with the software that it has been used, are given only those that we used.
I would add this to the limitation, I would also look into the software to see if you can export the raw data out and manually analyse as you are leaving a lot on the table that would be of interest.
Authors Response: Thanks for this suggestion! The authors completely agree with reviewer´s proposal to be added in the limitations as more variables could have been evaluated. Now it is stated as the following: “… and more thorough CMJ analysis with kinetic and kinematic variables may demonstrate differences when compared between the days and groups.”
Reviewer 2 Report
Comments and Suggestions for Authors
Dear Authors,
All the revisions have been carefully answered, so I recommend accepting the article in its present form.
Congratulations.
Author Response
The authors really appreciate the reviewer’s time and effort to review the manuscript and reviewer´s suggestions and comments increased the quality of the paper.
Thanks in advance for the contribution on this process.